# Morphophysiological Assessment of the Cervix during the Reproductive Cycle and Early Pregnancy in Does Using Computed Tomography and Oxytocin Receptor Immunohistochemistry

**DOI:** 10.3390/ani14192793

**Published:** 2024-09-27

**Authors:** Supapit Kanthawat, Kongkiat Srisuwatanasagul, Fueangrat Thatsanabunjong, Nardtiwa Chaivoravitsakul, Saritvich Panyaboriban, Sayamon Srisuwatanasagul

**Affiliations:** 1Veterinary Bioscience Unit, Department of Physiology, Faculty of Veterinary Science, Chulalongkorn University, Bangkok 10330, Thailand; 6175514431@student.chula.ac.th (S.K.); 6175521831@student.chula.ac.th (F.T.); 2Department of Anatomy, Faculty of Veterinary Science, Chulalongkorn University, Bangkok 10330, Thailand; kongkiat.s@chula.ac.th; 3Diagnostic Imaging Unit, Small Animal Teaching Hospital, Faculty of Veterinary Science, Chulalongkorn University, Bangkok 10330, Thailand; nardtiwa.c@chula.ac.th; 4Faculty of Veterinary Science, Prince of Songkla University, Songkhla 90110, Thailand

**Keywords:** computed tomography, doe, morphophysiological assessment, oxytocin receptor

## Abstract

**Simple Summary:**

This study investigates the structural and functional changes in the cervix of doe goats during different reproductive stages using advanced imaging techniques and biochemical markers. By analyzing the cervix at various stages of the estrous cycle and early pregnancy, the study aims to provide insights into reproductive management strategies. The results highlight the dynamic changes in cervical morphology and oxytocin receptor expression, which are crucial for reproductive success in goats. This research may contribute to improve artificial insemination techniques and overall reproductive health in veterinary practice.

**Abstract:**

This study aimed to elucidate the morphophysiology and oxytocin receptor (OXTR) expression in the cervix of doe goats during various reproductive stages to enhance reproductive management strategies. A total of 40 cervical samples were categorized into follicular (*n* = 15), luteal (*n* = 10), and early pregnancy (*n* = 15) stages. Utilizing advanced imaging based on functional and morphological markers, the study employed computed tomography (CT) scans, histochemical staining (Masson trichrome and alcian blue), immunohistochemistry, Western blotting, and quantitative PCR (qPCR) to assess structural changes in the cervix and in OXTR expression during the estrous cycle and early pregnancy. CT scans revealed consistent cervical folds and a significant reduction in cervical width during pregnancy, suggesting structural adaptations for gestational integrity. Histochemical analyses indicated a well-organized collagen network and presence of mucins, essential for cervical function and integrity. Immunohistochemistry and Western blotting demonstrated elevated OXTR protein levels during the follicular stage, which were markedly reduced during pregnancy, indicating a role in facilitating cervical relaxation and sperm transport during estrus and maintaining cervical closure during gestation. qPCR analysis showed stable *OXTR* mRNA levels during follicular and luteal stages with a slight, non-significant increase during pregnancy, pointing towards posttranscriptional regulatory mechanisms. In conclusion, this study demonstrates that cervical morphology and OXTR expression in doe goats undergo significant changes across reproductive stages, with elevated OXTR protein levels during the follicular phase and notable reductions in cervical width and OXTR protein levels during pregnancy, indicating structural and functional adaptations for both reproductive processes and gestational integrity.

## 1. Introduction

The cervical canal, which serves as the connection between the uterus and vagina, plays a crucial role in doe reproductive biology as it is involved in multiple reproductive events such as during the follicular phase, when cervical mucus becomes permeable, allowing the facilitation of sperm transport to the uterus. Additionally, the cervix acts as a physical barrier, preventing the ascension of external pathogens, such as bacteria and other microorganisms, that could compromise uterine health, especially during other phases of the estrous cycle when mucus is less penetrable [1]. The cervix of small ruminants is an elongated fibromuscular canal characterized by numerous tissue folds or rings, ending in an external cervical os exhibiting diverse forms [2,3]. During estrus, the sheep cervix undergoes collagen remodeling, by periovulatory hormone profiles, that leads to degradation of collagen fibers and cervical relaxation and dilation [4,5,6]. This structural change is widely used for transcervical introduction of artificial insemination (AI) and embryo transfer tools associated with assisted reproductive techniques (ARTs) in ruminants.

The use of computed tomography (CT) in the study of the goat cervix represents a significant advancement in veterinary imaging, offering detailed cross-sectional views crucial for reproductive research and clinical practice. CT imaging enables non-invasive examination of structural and morphological changes in the goat cervix across different reproductive states, such as the estrous cycle and pregnancy, as well as in response to various treatments, thereby enhancing both research and clinical applications as previously applied in other species [7].

The morphology of the ewe cervix demonstrates that the closure and opening of the cervical canals are controlled by several hormones and biological factors during the estrous cycle [3]. Many studies have examined the morphological and physiological characteristics of the ewe cervix, but the caprine cervix remains underexplored, requiring further research to understand its distinct structural and functional dynamics. Cervical relaxation, primarily regulated by hormonal changes and receptor dynamics, has been extensively studied in relation to parturition in ruminant species [8], including goats [9]. This process involves significant structural modifications in the cervix, such as dilation, relaxation, and ripening, driven largely by collagen remodeling and degradation [10,11,12]. These changes increase cervical compliance, facilitating crucial reproductive events like sperm migration and parturition. Further research focused on the caprine cervix is necessary to clarify the specific mechanisms underlying its structural changes during different reproductive stages, contributing to a more detailed understanding of ruminant reproductive physiology. Moreover, the role of oxytocin receptors in the cervical canal is of particular interest. The binding of oxytocin to its receptors can induce softening or contractions depending on the stage of the reproductive cycle in ruminants. Furthermore, oxytocin is involved in cervical ripening during parturition and is known to cause cervical relaxation and dilation [10]. These observations suggest that oxytocin not only facilitates parturition by softening the cervix and enhancing uterine contractions but may also be involved in the cervical structural changes that happen during the estrous cycle in goats.

New insights into the morphology of the cervical canal and the dynamics of oxytocin receptor expression during the estrous cycle and early pregnancy in goats are crucial for better understanding the mechanisms regulating cervical function in ruminants. Such information can contribute to improving the efficiency of assisted reproductive techniques (ARTs) and reproductive interventions in goats and related species. Therefore, this study aimed to characterize the cervical morphophysiology using CT scanning, histochemical staining, and oxytocin receptor (OXTR) expression analysis across different reproductive stages in doe goats.

## 2. Materials and Methods

### 2.1. Samples

The procedures and animals used were approved by Chulalongkorn University Laboratory Animal Center Animal Care and Use Protocol (CULAC-ACUP); protocol number 2031095. The whole female reproductive tracts were collected from Boer and crossbred goats at a certified slaughterhouse in Bangkok, Thailand. Samples were categorized into different reproductive stages: follicular (*n* = 15), luteal (*n* = 10), and early pregnancy stage (*n* = 15). The confirmation of each sample’s stage relied on the presence or absence of the corpus luteum or preovulatory follicles on the ovaries. Samples with the presence of the corpus luteum on any ovaries were considered as at the luteal stage. Samples with growing follicles (4–9 mm in diameter) and regressed corpus luteum were considered as at the follicular stage [1,13]. In the early stage of pregnancy, samples included were within their first trimester (up to 7 weeks’ gestation) [14], with gestational age determined by measuring the crown–rump length (CRL) of the fetus. In the present study, the CRL of the fetus ranged from 25.8 to 44.3 mm, estimating the pregnancy age from 5 to 7 weeks [15,16,17].

### 2.2. Computerized Tomography Scans (CT Scans)

CT scans of the doe reproductive tracts were performed by a helical CT unit (Optima CT660^®^, GE, Bangkok, Thailand) at the diagnostic imaging unit, Small Animal Teaching Hospital, Faculty of Veterinary Science, Chulalongkorn University, Thailand. To examine the real appearance of the cervical canal, at least 5–10 mL of diluted contrast medium, iohexol (1:20, OMNIPAQUE™, Cork, Ireland), was inserted into the cervical canal until a small leak could be visible at the external os. The images of each specimen were recorded in Digital Imaging and Communications in Medicine (DICOM) files and analyzed using Horos^®^ software (Version 3, Geneva, Switzerland) (Figure 1). In detail, using 3D MPR viewing mode, the canal length measurement was conducted in three different views which were dorsal, lateral, and posterior views. The result of the average from all plane views is shown in diameter (mm). If the contrast medium could not pass through the canal or was not visible due to blockage from the cervical folds, it was measured as zero in diameter.

### 2.3. Morphological Data

Anatomical measurement was performed after the samples were evaluated with a CT scan. The appearance of the external os was classified based on a classification system developed in ewes by Kershaw et al. [3] (Figure 2). The cervix was cut longitudinally from the external os to the uterine body (Figure 1). The following data were collected: number of cervical folds, cervix length (from external os to uterine body opening), degree of completeness, and the interdigitation of the cervix, as described by Kershaw et al. [3] (Figure 3). Briefly, the cervical canal pathways were divided into three grades as followed: grade 1 showed a straight canal with inter-cervical fold space, making a fishbone-like shape (Figure 3A,D). The longitudinal section showed completed and aligned cervical folds (Figure 3G). Grade 2 had a crooked canal around the second or third fold, with a normal straight canal after the crookedness, corresponding to the incompletion of cervical folds at the level of the crooked canal as shown in the longitudinal section (Figure 3B,E,H). In grade 3, the canal was long, narrow, and crooked, and inter-cervical fold could not be distinguished (Figure 3C,F). The longitudinal section showed incomplete and unaligned cervical folds (Figure 3I).

### 2.4. Masson Trichrome and Alcian Blue Staining

Masson trichrome was used to stain the connective tissue fiber in cervical sections. The cervical tissues, approximately 10 × 5 × 5 mm^3^, were primarily collected from the midsection of the cervix in each sample. Following histological processing, the cervical sections were deparaffinized and rehydrated using a series of graded alcohols. The Masson trichrome staining was performed according to a previous protocol in canine intestinal tissue [18]. In brief, the cervical tissues were treated in preheated (58 °C) Bouin’s solution for 60 min to enhance tissue preservation and staining and then washed thoroughly until the yellow color disappeared. The nucleus was stained with Weigert’s iron hematoxylin for 5 min and differentiated in 0.5% hydrochloric acid in 70% alcohol for 5 s. The sections were stained with Biebrich’s scarlet acid solution, phosphomolybdic acid solution, and aniline blue, respectively, with 3X thorough washing with distilled water between each solution. Sections were dehydrated in graded alcohol before being mounted with mounting medium (Permount^®^, Fisher Chemical, Hampton, NH, USA). 

For Alcian blue staining, an Alcian blue staining kit (pH 2.5, Vector Laboratories Inc., Burlingame, CA, USA) was used. The cervical tissue sections were consecutively cut from the same block used for Masson trichrome staining. Briefly, after deparaffinization, sections were sensitized by acetic acid solution, followed by Alcian blue staining for 30 min, and excess staining was washed with acetic acid and water. Then, sections were counterstained with Nuclear Fast Red stain for 5 min and washed with water. 

All stained sections were scanned with a digital slide scanner (3DHISTECH, Budapest, Hungary). The percentages of collagen and mucin from Masson trichrome and Alcian blue staining, respectively, were analyzed using the image analysis software CellQuant (3DHISTECH QuantCenter, Budapest, Hungary). In detail, the regions used for analysis were carefully chosen based on their histological integrity. For both the epithelial and stromal compartments, annotation areas were defined as squares measuring approximately 0.25 mm^2^. These uniformly sized regions allowed for consistent assessment across all samples. The areas selected for annotation were free of artifacts, necrosis, or poorly stained regions to avoid skewed results. Multiple non-overlapping regions were chosen from each cervical compartment to ensure representativity of the entire tissue section. Thereafter, image analysis software was used to precisely calculate the percentage of stained areas within each selected region.

### 2.5. Oxytocin Receptor Immunohistochemistry

The cervical samples were fixed in 4% paraformaldehyde for 48 h, transferred to 70% alcohol, processed using an automatic tissue processor (Tissue-Tek VIP 5 Jr., Sakura, Tokyo, Japan), and embedded in paraffin. The paraffin blocks were cut into 4 µm thick sections using a microtome (RM 2035, Leica Biosystems Nussloch GmbH, Nussloch, Germany). Section slides were warmed up at 60 °C for 24 h prior to staining to remove excess water. The tissue sections were deparaffinized in xylene and rehydrated through graded alcohol. The oxytocin receptor immunohistochemical protocol was adapted from earlier studies in canine tissues [19,20]. For the antigen retrieval method, the tissue sections were placed in 0.01 M citrate buffer (pH 6.0) and heated in a microwave oven at 750 W for 10 min (2 times × 5 min). Endogenous peroxidase was blocked using freshly prepared 3% hydrogen peroxide (H_2_O_2_) in methanol for 10 min. Afterward, slides were incubated with normal serum (Vector Laboratories, Burlingame, CA, USA) to reduce non-specific staining reactions. The rabbit polyclonal anti-oxytocin receptor (1:200, bs-1314R, Bioss Antibodies, Boston, MA, USA) was used as primary antibody. After overnight incubation of primary antibody, the slides were incubated with HRP-conjugated secondary antibody (1:200, Cell Signaling Technology, Danvers, MA, USA) for 30 min at room temperature. In the final step, nova red (Vector NovaRED^®^, Vector Laboratories, Burlingame, CA, USA) was applied as a chromogen and counterstained with Mayer’s hematoxylin afterward. The uterine samples which were known to express oxytocin receptors were used as positive controls while the negative control was performed by replacement of the primary antibody with PBS. The stained slides were scanned using a digital slide scanner (3DHISTECH, Budapest, Hungary) and evaluated using an image analysis program (3DHISTECH, Budapest, Hungary). Immunoreactivity was analyzed by the image analysis software CellQuant (3DHISTECH QuantCenter, version 2.2.1.88915, Budapest, Hungary), and the result was calculated and shown as Histoscore (H-score). The H-score is a quantitative method used to assess immunoreactivity in tissue samples. It is calculated by evaluating both the intensity of staining and the percentage of cells showing that intensity. The positive staining intensity levels for OXTR were determined based on multiple stained slides to ensure accurate measurement of all positive levels. Once the calibration was finalized, the settings were saved to a file which was then applied to all subsequent slide evaluations. Finally, the H-score is determined by assigning a score to the staining intensity and multiplying it by the percentage of cells at each intensity level. By this calculation, higher scores indicate stronger and more widespread staining [21]. 

### 2.6. RNA Extraction, Reverse Transcription, and Real-Time qPCR

The RNA preparation protocol was adapted from an earlier study by Permkam et al. [22]. Total RNA was isolated from frozen does’ cervical tissues, four samples for each group, using a GenUP^™^ Total RNA Kit (Biotechrabbit GmbH, Berlin, Germany), and treated with Deoxyribonuclease I (Serva Electrophoresis GmbH, Heidelberg, Germany) to remove genomic DNA according to the manufacturer’s instructions. The RNA concentration and purity were accessed using a spectrophotometer at 260 and 280 nm (Nanadrop One, Wilmington, DE, USA). Subsequently, single-stranded complementary DNA (cDNA) was synthesized using SuperScript^®^ III First-Strand (Invitrogen, Thermo Fisher Scientific, MA, USA) according to the guidelines supplied by the manufacturer.

For quantification of mRNA expression of *OXTR*, the quantitative real-time PCR (qPCR) was performed. Primer pairs of OXTR (forward: GCCTTCATCGTGTGCTGGA, reverse: GAAAGCCGAGGCTTCCTTG, XM_018038465.1) were designed inferring from published caprine sequences. The target gene was amplified by a KAPA SYBR^®^ FAST Universal qPCR Kit (KAPA Biosystems, Wilmington, MA, USA), using a CFX96^™^ real-time PCR detection system (Bio-Rad, Hercules, CA, USA). The PCR reactions were set as 95 °C for 3 min, followed by 39 cycles of 95 °C for 10 s, 55 °C for 20 s, and 72 °C for 1 s. In this study, glyceraldehyde-3-phosphate dehydrogenase (GAPDH, forward: CACCCTCAAGATTGTCAGCAA, reverse: CGTGGACAGTGGTCATAAGT, XM_005680968.3) served as a reference gene. The mRNA expression level was estimated with the ΔCt method [23].

### 2.7. Western Blot Analysis

The Western blot analysis protocol was performed by using total protein normalization with stain-free technology [24]. In detail, frozen cervical tissues were ground to powder with a sterile mortar and pestle and treated with RIPA buffer (150 mM NaCl, 2 mM EDTA, 1% Triton X-100, 1 mM NaF, 50 mM Tris-HCl, pH 7.4) containing protein inhibitor (Protease inhibitor cocktail set I, EMD Millipore Corporation, Temecula, CA, USA). The sample was vortexed and centrifuged at 12,000 rpm at 4 °C for 20 min, after which supernatant was collected. Protein concentrations were determined by BCA Protein assay (Dual-Range™ BCA Protein Assay Kit, Visual Protein, Taipei, Taiwan). Protein extracts (20 μg) were warmed up to 95 °C for 5 min in loading buffer with β-mercaptomethanol and separated on a 10% SDS-PAGE (TGX Stain-Free™ Fast-cast™ acrylamide solutions, Bio-Rad, Hercules, CA, USA). Gel was activated by UV light for 5 min prior to transfer onto PVDF membranes (Immobilon-P, Millipore, Bedford, MA, USA); both steps were performed using a Bio-Rad system according to the manufacturer’s instructions. Protein membrane was blocked with 5% (*w*/*v*) non-fat dry milk in TBS-T (20 mM Tris, 500 mM NaCl, and 0.1% Tween 20) for 1 h at room temperature and incubated overnight at 4 °C with primary antibodies against OXTR (1:1000) as used in IHC. Horseradish peroxidase (HRP)-conjugated anti-rabbit IgG (cat. 7074P2, Cell Signaling Technology, Danvers, MA, USA) was used as secondary antibody at a dilution of 1:5000 in TBS-T and incubated at room temperature for 1 h with intensive washing in TBS-T between each step. The membranes captured the total protein with stain-free blot mode on a ChemiDoc MP (Bio-Rad, Hercules, CA, USA). The protein bands of interest were visualized using enhanced chemiluminescence (ECL; Immobilon Crescendo Western HRP Substrate, Millipore, Burlington, MA, USA). The densitometry was quantified for the protein of interest and normalized to total protein signal using Image Lab 6.0 (Bio-Rad, Hercules, CA, USA).

### 2.8. Statistical Analysis

All statistical analyses were conducted using SPSS version 29.0.0.0. Unless specified otherwise, all numerical results are presented as the mean ± standard error of the mean (SEM). Data normality was assessed using the Kolmogorov–Smirnov and Shapiro–Wilk tests to determine the appropriate analytical methods. A one-way ANOVA was performed to compare the means of multiple groups, and when significant differences were found, the least significant difference (LSD) post hoc test was applied to identify specific group differences. A significance level of 0.05 was used for all statistical tests.

## 3. Results

### 3.1. Cervical Canal Morphometric Data, Gross Appearance, and CT Imaging

The morphometric analysis of the doe goat cervix based on CT scans revealed that the average number of cervical folds was five, with a range of three to seven rings (Table 1). The cervical length remained consistent across different reproductive stages. Cervical grades were distributed as follows: 45.0% (18) of cervices were classified as grade 1, 35% (14) as grade 2, and 20% (8) as grade 3. Notably, the cervical diameter of early pregnant does was significantly smaller compared to those in the follicular and luteal stages (*p* < 0.05). Furthermore, the average size of the cervical canal varied significantly among the different stages (*p* < 0.05). Despite these differences in cervical size, there was no significant correlation between cervical grade and estrous stage; each stage displayed a consistent pattern with the majority of cervices classified as grade 1 and the fewest classified as grade 3.

Gross examination of the goat cervix revealed prominent cervical rings that provide rigidity and define the structure of the cervix. These rings encircle the cervical canal, which serves as a central passage connecting the uterine body to the vagina (Figure 1). The external os of the cervix was categorized into five distinct types (Figure 2). The most common types observed were the rose (10, 25%) and the flap (10, 25%), followed by the duckbill (9, 22.5%), the slit (8, 20%), and the papilla (3, 7.5%).

Computed tomography (CT) scans of the doe goat cervix, particularly in the inverted mode, highlighted the cervical rings as distinct, less dense structures against a darker background. This imaging modality made the central cervical canal discernible as a darker passage within the cervix, surrounded by comparatively lighter muscular layers. These layers contrasted sharply with the cervical rings and the canal, allowing for clear visualization of the cervix’s anatomical boundaries and the integration of its internal structures within the reproductive system (Figure 1B). Additionally, 3D figures of the cervix were generated using the 3D-MIP volume rendering function, illustrating the gross appearance of the cervical canal according to different cervical grades (Figure 3). However, it was observed that the cervices of pregnant does were tightly closed, preventing the contrast medium from filling the cervical canal and, thus, the 3D volume rendering of pregnant cervices could not be visualized.

### 3.2. Masson Trichrome Staining

The percentages of collagen fiber, as determined by Masson trichrome staining at different estrous stages, are summarized in Table 2. Significant differences in the percentage of collagen fiber were observed among the stages in the muscular layer (Figure 4). Notably, the collagen content in the muscular layer did not differ between the follicular and luteal stages but it was markedly highest in the pregnant cervix (*p* < 0.05). Furthermore, histological examination revealed that collagen fibers in the follicular stage were slightly more detached from each other compared to the luteal stage (Figure 4).

### 3.3. Alcian Blue Staining 

Alcian blue staining was utilized to identify acidic sulfate mucosubstances, hyaluronic acid, and sialomucin. The percentage of positively stained cells within the epithelial layer is summarized in Table 2. The blue staining indicated positive results, predominantly observed in the cervical surface epithelium, as illustrated in Figure 5. Although there was no significant variation in positive staining between non-pregnant and pregnant states, consistent positive staining patches were detected in the cervical secretions of pregnant samples (Figure 5C), suggesting the presence of mucin in the cervical plug during pregnancy. In cervical samples collected postovulation, positive staining was not restricted to the apical surface, as observed in the follicular stage, but was also evident in goblet cells throughout the cervical epithelium (Figure 5A).

### 3.4. Immunohistochemistry

#### 3.4.1. OXTR Expression in Cervix Tissue 

The immunolocalization of OXTR was observed in the cytoplasm of the cervical epithelial cells, connective tissue of the subepithelial tissue layer, muscular layer, and tunica media (smooth muscle layer) of the vessels as shown in Figure 6. In addition, the semiquantitative results presented as H-score for various cervical compartments at different reproductive stages are demonstrated in Figure 7. During the estrous cycle, the OXTR H-score expression was significantly higher in all compartments during the follicular stage compared to early pregnancy. Furthermore, when comparing between follicular and luteal stages, significantly higher OXTR expression was demonstrated in the subepithelial layer, muscular layer, and smooth muscle of cervical vessels (*p* < 0.05) (Figure 7). Higher OXTR H-score in all compartments was noticeable during non-pregnant stages and it declined during early pregnancy.

#### 3.4.2. OXTR Protein Expression by Western Blot

The results demonstrated the presence of a 43 kDa protein band corresponding to OXTR in the doe goat cervices. The mean relative expression levels of OXTR protein (mean ± SEM) were 53.79 ± 7.19% in the follicular stage, 30.78 ± 11.59% in the luteal stage, and 23.87 ± 0.89% in the pregnant stage. The relative expression of OXTR protein in the follicular stage was significantly higher than in the pregnant stage (*p* < 0.05). Furthermore, the luteal stage exhibited a slightly lower relative OXTR expression compared to the follicular stage though not significantly different (Figure 8).

#### 3.4.3. *OXTR* mRNA Expression

The *OXTR* mRNA expressions in relation to GAPDH were investigated in the doe cervix by RT-qPCR. The results showed no difference in relative mRNA expression between the follicular and luteal stages. The mean (±SEM) of mRNA expression of *OXTR* in relation to GAPDH was 1.54 ± 0.79, 1.43 ± 0.09, and 4.07 ± 2.87 during the follicular, luteal, and pregnancy stages, respectively. The pregnant group exhibited a slight increase in *OXTR* mRNA levels, with values 2.64 and 2.84 times greater than those observed in the follicular and luteal stages, respectively. However, this elevation did not reach statistical significance.

## 4. Discussion

The morphometric data from CT scans show relatively consistent cervical folds during different physiological estrous stages. This suggested that the structural complexity of the cervix, indicated by the number of folds, did not change significantly during different reproductive stages. There was a notable reduction in cervix width from the non-pregnant to pregnant stages while both the follicular and luteal stages have similar cervix width. This substantial reduction in cervix width is likely related to physiological preparations for pregnancy maintenance, during which the cervix remodeled to protect the embryo and became more flexible for delivery in the later stages of pregnancy. Regarding cervical length, it remained relatively stable during different stages, with only a slightly greater length observed during the luteal stage compared to the follicular stage and pregnancy. This stability in length indicated that, despite changes in cervical size, its overall length remained constant. Conversely, in pregnant women, the cervix decreased in length and increased in width from mid-pregnancy to term [25]. This inconsistency might be associated with the site of embryo implantation. In humans, the embryo implants in the uterine body, which is relatively close to the cervix. In doe goats (and many other ruminants), embryonic implantation occurs in the uterine horns, which are farther from the cervix than in humans. This anatomical difference necessitates a reduction in cervical width to effectively close off the passage to external factors that could be harmful to the developing embryo. In addition to the length and width of the cervix, the size of the cervical canal exhibits significant variation during different stages and positions of the cervical folds. During the follicular phase, the cervical canal was relatively enlarged, whereas a reduction in size was observed during the luteal phase. The most pronounced decrease occurs during early pregnancy, indicating a marked constriction of the cervical canal. This constriction is crucial for maintaining pregnancy, as a closed cervix helps to prevent infections and support the developing fetus. Furthermore, no correlation between the types of external os and cervical morphometric data was found in doe goats from the present study. This was consistent with previous findings using silicone molds in sheep which reported the irrelevance of the cervical canal morphology to the external os of the cervix [26]. Thus, the changes in the cervical morphometry are likely relate to the reproductive stages and pregnancy, highlighting the cervix’s dynamic response to hormonal and physiological needs during these periods. In sheep, it was demonstrated that complexity of the sheep cervix depends not only on physiological status but also breed, age, and parity [27]. However, the study of cervical morphology in doe goats remains limited, and this lack of detailed morphometric data may contribute to the lower success rates observed in transcervical insemination (TCI) compared to cattle. While artificial insemination (AI) efficiency in goats was generally comparable to cattle, the specific challenges associated with TCI in goats may be due to anatomical and morphological differences in the cervix.

The application of Masson trichrome and Alcian blue staining has provided valuable insights into the biochemical and structural properties of the cervix, which undergoes substantial changes throughout the reproductive cycle. Masson trichrome staining revealed a well-organized arrangement of collagen fibers in the goat cervix, prominently stained in blue, indicating a robust connective tissue framework. The orientation and density of these collagen fibers are critical in maintaining the cervix’s biomechanical properties, which support its reproductive function by allowing changes in tensile strength and extensibility during different stages of the reproductive cycle [4]. The integrity and organization of collagen fibers are crucial for cervical softening and dilation, processes regulated primarily by hormonal signals, particularly estrogen [4]. In ruminants, estrogen modulates the extracellular matrix (ECM), leading to the reorganization of collagen fibers, which allows the cervix to transition from a rigid, closed state to a more pliable structure [8]. This transformation is vital for facilitating sperm passage during the follicular phase and for enabling cervical dilation during parturition [28]. The observed decrease in collagen density during the follicular phase, compared to early pregnancy, from the present study results in reduced tensile strength, rendering the cervical tissue more pliable and capable of expanding to facilitate sperm passage to the uterus. In addition to collagen fibers, the presence of smooth muscle fibers, which are stained red with Masson trichrome, is evident throughout the cervical tissue. These fibers are integral to the cervical muscular composition, playing a vital role in its contractile function, which is necessary for both maintaining cervical integrity during pregnancy and facilitating dilation during labor [29]. Furthermore, the interplay between collagen and other extracellular matrix components, such as glycosaminoglycans, further contributes to the cervix’s ability to undergo significant mechanical changes without compromising tissue integrity. This dynamic remodeling process is essential not only for normal parturition but also for the prevention of conditions such as cervical insufficiency, which can lead to preterm birth [30]. The increased collagen content during pregnancy, as revealed by Masson trichrome staining, likely contributed to the increased tensile strength of the cervix, which is crucial for maintaining pregnancy as well.

Besides Masson trichrome staining, the utilization of Alcian blue staining to identify acidic sulfate mucosubstances, hyaluronic acid, and sialomucin in the cervix yielded significant insights into the distribution and presence of mucins at different reproductive physiological stages. The predominant blue staining observed in the cervical surface epithelium underscores the substantial presence of these mucosubstances in this compartment, aligning with their roles in maintaining mucosal integrity and function [31,32]. During follicular stage, the presence of acidic sulfate mucosubstances and sialomucin in cervical mucus, as highlighted by Alcian blue staining, suggested a role in creating a protective barrier and facilitating sperm migration from the cervix to the oviductal sperm reservoir [31,32,33,34]. In addition to the follicular stage, the detection of positive staining patches within the cervical secretions of pregnant samples indicated a marked increase in mucin content within the cervical plug, which was likely relevant for its role in protecting the uterine environment during pregnancy. This finding supported the hypothesis that the cervical plug’s composition is dynamically regulated to enhance its barrier properties [35]. Furthermore, the variation in staining patterns, where positive staining was found prominently in goblet cells, suggested a hormonally mediated alteration in mucin distribution and secretion. During the follicular phase, under the influence of estrogen, mucin secretion is notably in the goblet cells. This heightened mucin activity supports previous findings that spermatozoa are rapidly separated from seminal plasma upon entering the cervical mucus postmating [34]. Conversely, in the luteal phase, characterized by elevated progesterone levels, mucin production becomes more viscous [36]. This increase in viscosity plays a crucial role in modulating the cervical environment, effectively creating a barrier that limits sperm penetration and shields the uterus from external influences, such as pathogens. Despite this, the cervical mucus retains sufficient permeability to allow for the movement of immune cells and the exchange of nutrients. The differential localization and altered viscosity of mucin between these phases underscore the complex regulatory mechanisms driven by hormonal shifts, which precisely control mucosal composition and function in response to the changing requirements of the reproductive cycle. 

Regarding the study of OXTR protein, the analyses of OXTR protein expression in the goat cervix in the present study by immunohistochemistry and Western blot techniques have shown similar patterns. The ability to localize and visualize OXTR within tissue architecture by immunohistochemistry complemented the quantitative precision of Western blotting, providing a clear picture of OXTR dynamics in the goat cervix during different reproductive stages. The study of ewes demonstrated that the peak cervical relaxation and penetration by artificial insemination pipettes coincided with elevated levels of LH and estradiol, suggesting that the mechanism of cervical relaxation likely involves the oxytocin and prostaglandin pathways [5,37]. The results of oxytocin receptor expression by the immunohistochemical analysis presented as H-score revealed dynamic patterns of OXTR which were notably higher during the follicular stage in all cervical compartments. Conversely, the decrease in OXTR expression H-score during pregnancy might reflect a regulatory mechanism aimed at reducing oxytocin sensitivity in the cervix. Furthermore, the prominent OXTR expression in the vascular smooth muscle of the cervix may suggest a role in modulating blood flow during the reproductive cycle, even though the fibroelastic structure of the cervix limits significant muscular activity. In ruminants, PGE2 is a key mediator of cervical relaxation, and oxytocin may act as one of several factors that activate this pathway rather than directly influencing cervical contractility. Oxytocin’s vasodilatory effects, likely mediated through the release of endothelial nitric oxide [38], could enhance tissue perfusion and support cervical remodeling during pregnancy, consistent with the current understanding of cervical function in ruminants. In the cervix, increased OXTR expression in vascular smooth muscle could facilitate enhanced blood flow during times of peak reproductive activity, such as the follicular phase when preparation for sperm migration is critical. This enhanced blood flow could support the cervical tissue in facilitating an optimal immune response, which is crucial for eliminating microbial contamination. On the other hand, the reduction of OXTR expression in vascular smooth muscle during pregnancy could be part of the lower cervical contractility as lower levels of OXTR may help stabilize the cervical tissue by reducing the risk of unnecessary contractions which may lead to cervical incompetence or abortion. This stabilization was crucial to maintain the integrity of the cervical barrier throughout the gestation period [39]. In the cervical muscular layer, the higher H-score observed during the follicular phase, compared to the luteal and early pregnancy stages, suggested a distinct physiological adaptation associated with reproductive processes. The higher OXTR during the follicular phase may facilitate cervical relaxation and dilation, essential for successful sperm migration. The decline in OXTR expression during the luteal and early pregnancy stages compared to during the follicular stage may contribute to re-establishment of cervical closure. This pattern of OXTR expression underscores the dual role of the cervical muscular layer in both facilitating conception during the follicular phase and subsequently maintaining cervical integrity in the other stages.

For the Western blot result, OXTR expression was significantly higher during the follicular phase, underscoring oxytocin’s primary role in cervical relaxation and softening. These findings aligned with the immunohistochemistry results, indicating that oxytocin enhanced cervical receptivity and relaxation, promoting sperm transport. In the luteal phase, OXTR levels by Western blotting were lower when compared to the follicular stage, although the difference was not statistically significant. The reduced expression in the luteal phase compared to the follicular phase may imply a decreased need for cervical changes and a focus on maintaining uterine quiescence. The statistically significant difference in OXTR expression between the follicular and early pregnant stages reflected the cervix’s shift towards a protective function. This downregulation was likely influenced by progesterone dominance, which promotes cervical closure and the formation of the cervical plug, essential for maintaining uterine integrity and protecting the developing embryo from external pathogens. The lower OXTR levels reduced the cervix’s sensitivity to oxytocin, thus preserving its firm structure and ensuring the stability of early pregnancy. 

The quantitative PCR (qPCR) analysis of *OXTR* mRNA expression in the goat cervix revealed intriguing insights into the regulation of this gene during different stages of the reproductive cycle. The results indicated that there was no significant difference in relative mRNA expression of *OXTR* between the follicular and luteal stages, when normalized to GAPDH. This suggested a relatively stable transcriptional activity of the *OXTR* gene during these stages. In contrast, the pregnant group exhibited a slightly higher *OXTR* mRNA expression, although this increase was not statistically significant. When compared to the Western blot results, which show a significant decrease in OXTR protein levels during the pregnant stage, this discrepancy suggested posttranscriptional regulatory mechanisms may be influencing OXTR protein expression, such as mRNA stability, translation efficiency, or protein degradation. Posttranscriptional regulation is a well-documented phenomenon where various cellular processes can modulate the translation of mRNA into protein or affect protein stability. For instance, microRNAs (miRNAs) are known to bind to mRNA and either promote its degradation or inhibit its translation, which could lead to a reduction in protein levels without corresponding changes in mRNA levels [40,41]. Additionally, mRNA-binding proteins can influence the stability of mRNA or its efficiency in being translated into proteins, adding another layer of regulation. Protein degradation pathways, such as the ubiquitin–proteasome system, can also selectively degrade proteins, further contributing to discrepancies between mRNA and protein levels [42]. These findings highlight the complexity of gene regulation where mRNA levels may not always correlate directly with protein expression, underscoring the importance of analyzing both mRNA and protein levels to fully understand gene expression dynamics and their physiological implications in the reproductive cycle of the goat cervix.

## 5. Conclusions

In conclusion, the present study elucidated the morphophysiology and regulation of oxytocin receptor (OXTR) in the goat cervix during different reproductive stages using CT scanning, immunohistochemistry, Western blotting, and qPCR. CT scanning revealed consistent cervical folds and reduced cervical width during pregnancy. Histochemical staining highlighted the roles of collagen and mucins in cervical functions. While *OXTR* mRNA levels were stable in the follicular and luteal phases, protein expression varied significantly, with a marked reduction during pregnancy, indicating posttranscriptional regulation. These findings enhanced understanding of cervical physiology and may improve reproductive management and therapeutic interventions in the doe goat cervix.

## Figures and Tables

**Figure 1 animals-14-02793-f001:**
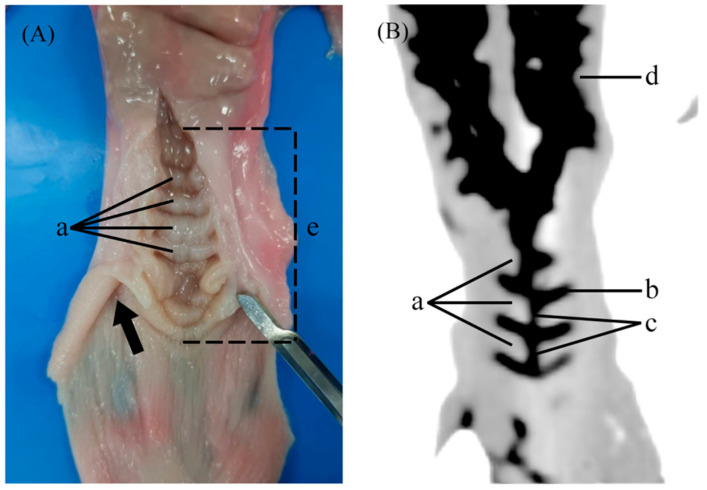
Longitudinal section of the cervix (**A**) and the cervical canal of doe goat obtained from CT scan in inverted mode (**B**) (contrast medium appears black). a, cervical folds; b, inter-cervical space; c, cervical canal; d, uterus; e, cervical length, length from external os to uterine body opening. Arrow indicates fornix.

**Figure 2 animals-14-02793-f002:**
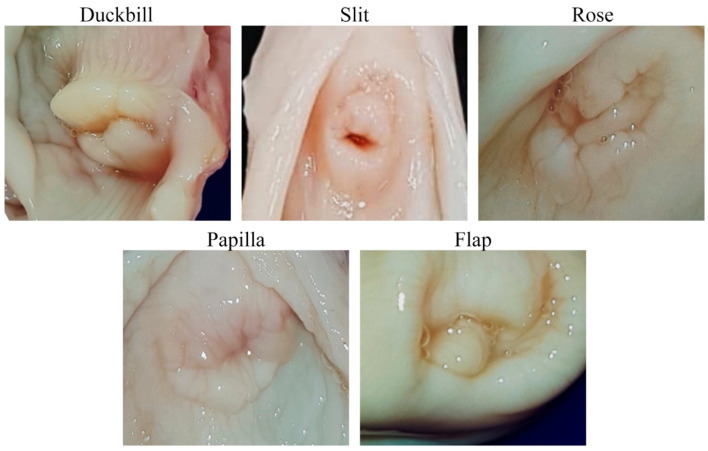
Morphological variations of the external os of the cervix in doe goats. The images depict five distinct morphological types of the external cervical os observed in doe goats: Duckbill, Slit, Rose, Papilla, and Flap. The Duckbill morphology shows a protruding, rounded shape, the Slit displays a narrow, elongated opening, the Rose exhibits a complex, folded structure, the Papilla is characterized by a papillary projection, and the Flap features a distinct flap-like formation.

**Figure 3 animals-14-02793-f003:**
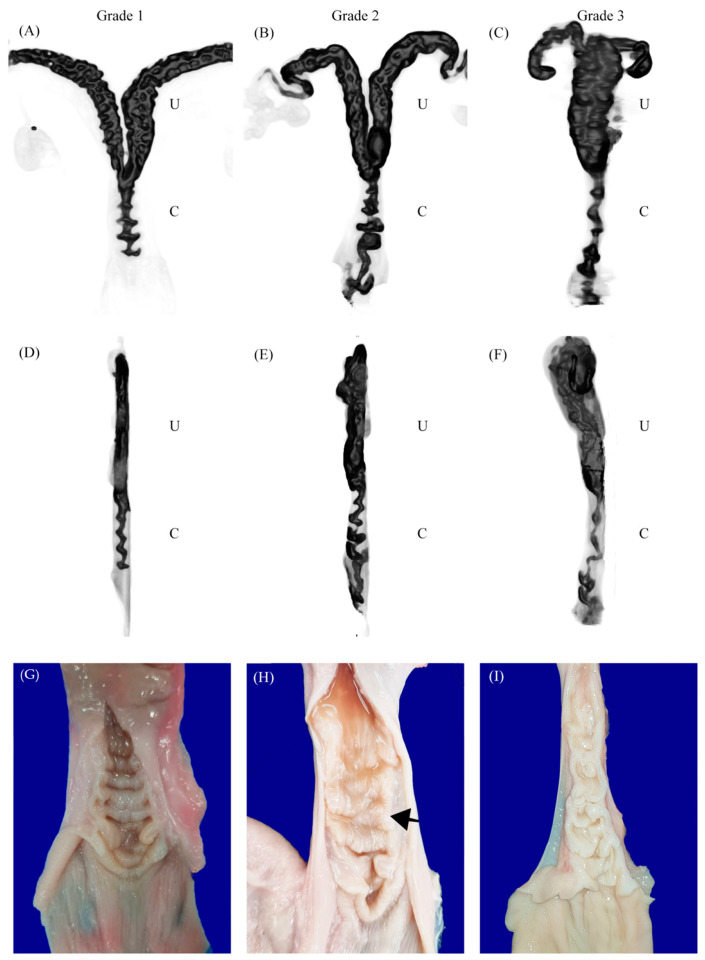
The 3D volume rendering of the cervical canal in two different views; dorsal and lateral views. Pictures are shown in inverted mode. Cervical canal grade 1 (**A**,**D**), grade 2 (**B**,**E**), and grade 3 (**C**,**F**). Gross anatomy of each cervical grade; grade 1 (**G**), grade 2 (**H**), and grade 3 (**I**). The arrow indicates a crooked canal. The letter “U” represents uterus, and the letter “C” represents cervix.

**Figure 4 animals-14-02793-f004:**
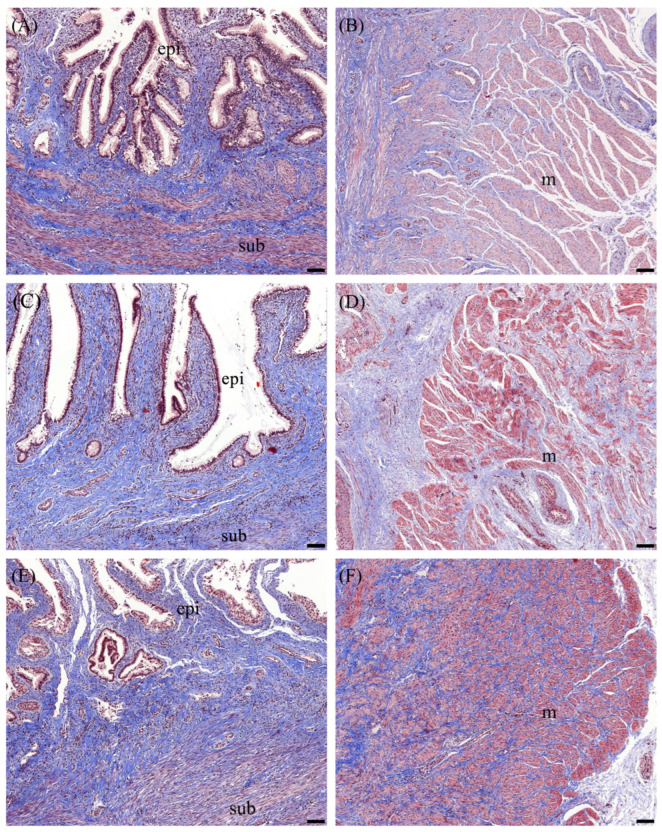
Masson trichrome staining of the epithelial and subepithelial layers (**A**,**C**,**E**) and the muscular layer (**B**,**D**,**F**) of the cervix during different reproductive stages. Panels (**A**,**B**) represent the follicular stage, panels (**C**,**D**) correspond to the luteal stage, and panels (**E**,**F**) show the pregnant stage. Collagen fibers and connective tissue are stained blue, muscle fibers appear red/pink, and epithelial cells are shown in purple/red. The abbreviation “epi” denotes the cervical epithelium, “sub” indicates the subepithelial layer, and “m” refers to the muscular layer. The scale bar represents 100 µm.

**Figure 5 animals-14-02793-f005:**
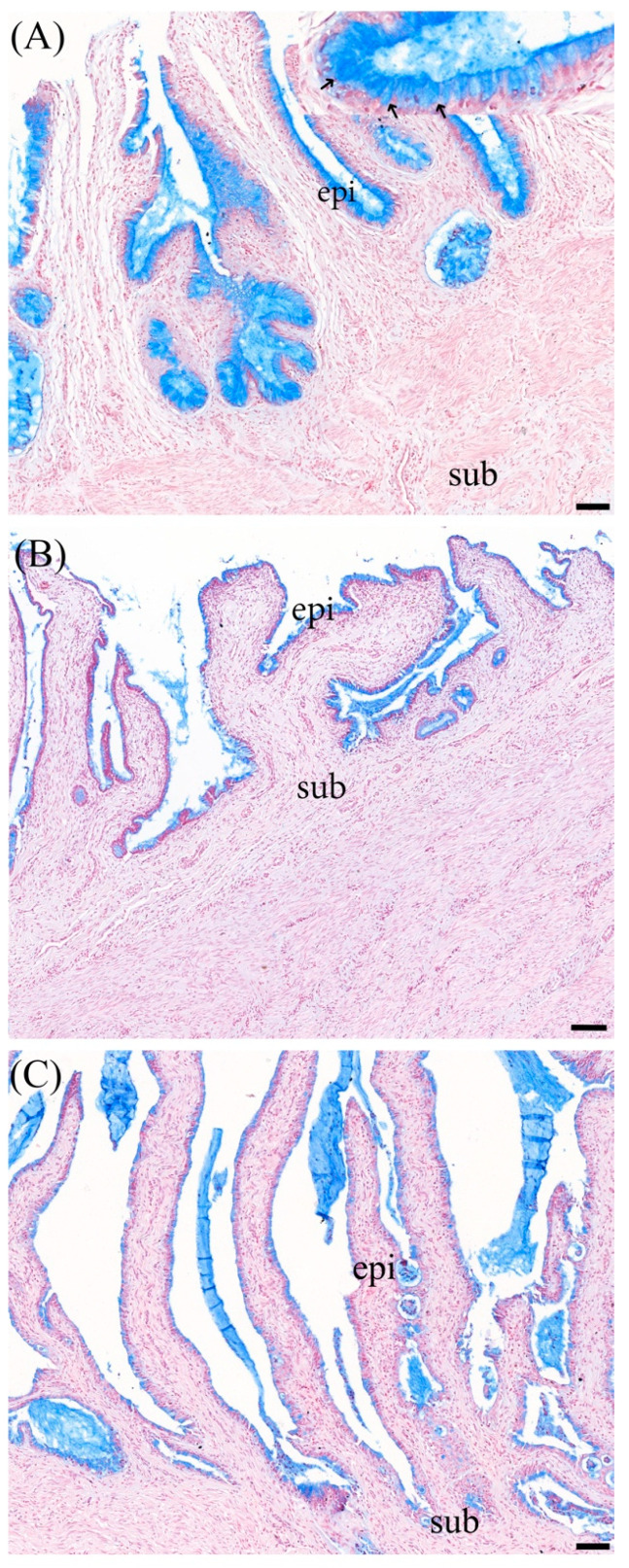
Alcian blue staining of doe cervical tissue during different reproductive stages: (**A**) follicular stage, (**B**) luteal stage, and (**C**) early pregnancy. Positive blue staining, indicating the presence of acidic mucosubstances, is observed in the epithelial layer of the cervix. Arrows point to goblet cells, which are responsible for mucin production. The abbreviation “epi” denotes the cervical epithelium, while “sub” indicates the subepithelial layer. The scale bar represents 100 µm.

**Figure 6 animals-14-02793-f006:**
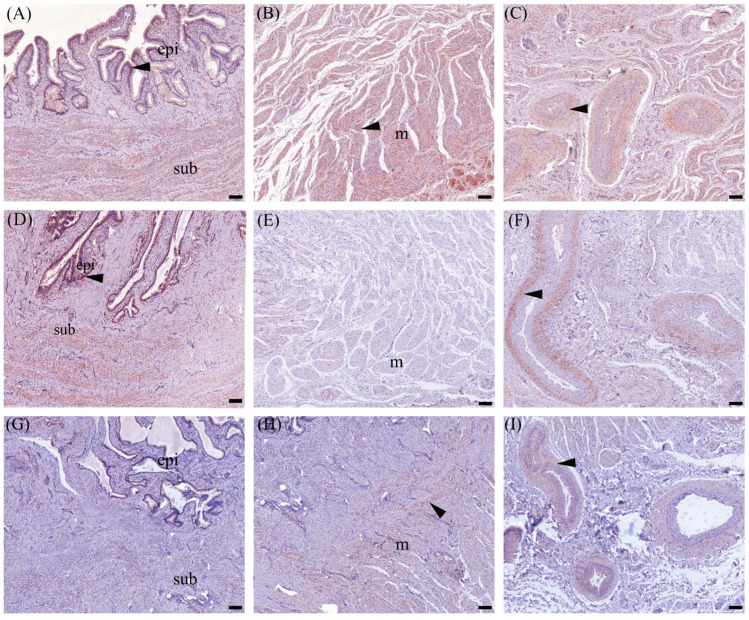
The immunolocalization (**A**–**I**) of the OXTR, shown as red-brown staining (nova red), in different regions of doe goat cervical tissue, epithelial and subepithelial layer (**A**,**D**,**G**), deep muscle layer (**B**,**E**,**H**), and around the vascular smooth muscle (**C**,**F**,**I**), at the follicular (**A**–**C**), luteal (**D**–**F**) stages of estrous cycle, and during pregnancy (**G**–**I**). Arrow heads represent positive OXTR immunostaining cells. The abbreviation “epi” denotes the cervical epithelium, “sub” indicates the subepithelial layer, and “m” refers to the muscular layer. The scale bar represents 100 µm.

**Figure 7 animals-14-02793-f007:**
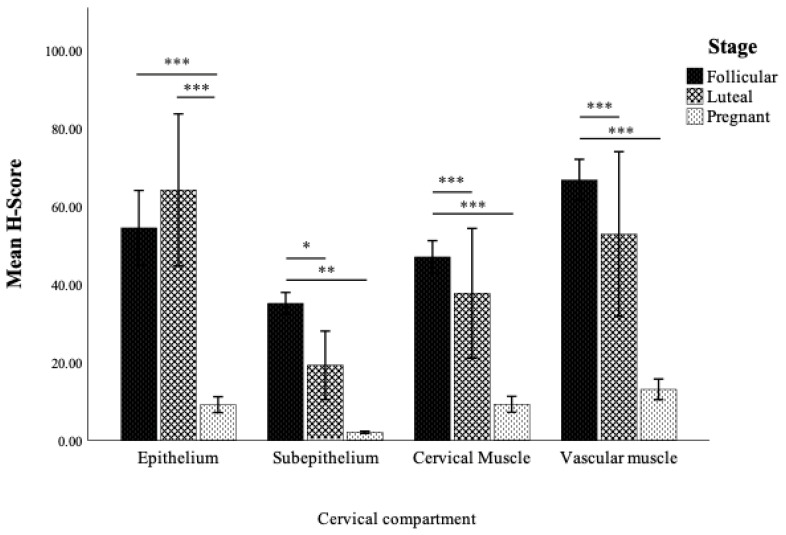
The immunohistochemical results of OXTR shown as H-score (mean ± SEM) in different compartments of the doe goat cervical tissue and during various reproductive stages. Statistical analyses were performed using one-way ANOVA with LSD post hoc paired comparison (* *p* < 0.05, ** *p* < 0.01, *** *p* < 0.001).

**Figure 8 animals-14-02793-f008:**
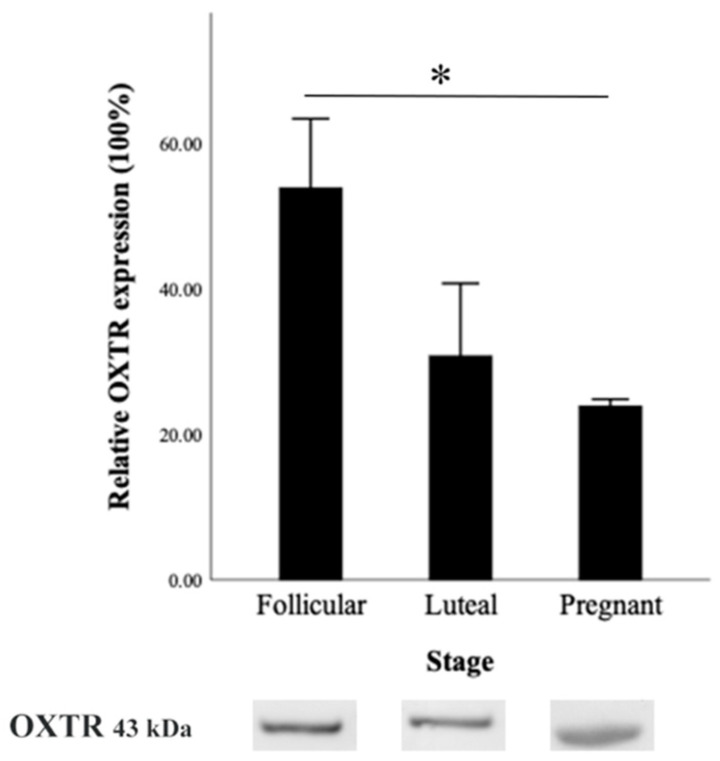
Relative OXTR expression in the cervical tissue of doe goats by Western blot during different reproductive stages. Representative Western blot images of OXTR (43 kDa) for each stage are shown below the graph. Data are presented as mean ± SEM. Asterisk (*) represents significant difference (*p* < 0.05).

**Table 1 animals-14-02793-t001:** Summary of the cervical canal morphometric data (mean ± SEM), cervical grade, and external os type.

Parameter		Reproductive Stage								
		Follicular Phase (*n* = 15)			Luteal Phase (*n* = 10)			Early Pregnancy (*n* = 15)		
Cervical folds (n)		5.13	±	0.26	5.50	±	0.31	5.20	±	0.26
Cervical width (mm)		14.42	±	0.66 ^a^	14.23	±	0.89 ^a^	11.08	±	0.70 ^b^
Cervical length (cm)		4.47	±	0.22	4.76	±	0.30	4.47	±	0.22
Cervical canal size										
	1st fold	1.82	±	0.11 ^a^	1.37	±	0.09 ^a,b^	0.36	±	0.20 ^b^
	2nd fold	1.81	±	0.14 ^a^	1.21	±	0.07 ^a,b^	0.32	±	0.17 ^b^
	3rd fold	1.71	±	0.13 ^a^	1.02	±	0.25 ^b^	0.11	±	0.12 ^c^
Average canal size		1.76	±	0.09 ^a^	1.23	±	0.10 ^b^	0.34	±	0.18 ^c^
Cervical grade										
	Grade 1		6			5			7	
	Grade 2		8			2			4	
	Grade 3		1			3			4	
External os type										
	Duckbill		2			4			3	
	Slit		2			3			3	
	Rose		3			1			6	
	Papilla		2			-			1	
	Flap		6			2			2	

a,b,c Within a row, means with different superscripts differed significantly at *p* < 0.05.

**Table 2 animals-14-02793-t002:** The histochemical analysis of the doe goat cervix at different reproductive stages using Masson trichrome and Alcian blue staining techniques. The data are expressed as the mean percentage (±SEM) of collagen fibers and Alcian-blue-positive staining in specific cervical compartments during the follicular, luteal, and early pregnancy.

Parameter		Reproductive Stage								
		Follicular Phase (*n* = 15)			Luteal Phase (*n* = 10)			Early Pregnancy (*n* = 15)		
Masson trichrome ^(1)^ Subepithelial layer										
	Collagen	38.94	±	1.98	36.52	±	2.20	36.12	±	1.87
Cervical muscle										
	Collagen	15.28	±	1.12 ^a^	13.14	±	1.16 ^a^	22.44	±	1.05 ^b^
Alcian blue stain ^(2)^										
	Epithelial layer	14.52	±	0.57	16.93	±	0.58	15.43	±	1.11

^(1)^ Percentage of collagen fiber in different compartments of doe goat cervix by Masson trichrome staining at different reproductive stages (mean ± SEM). ^(2)^ Percentage of Alcian-blue-positive staining in the epithelial layer of the cervix at different reproductive stages (mean ± SEM). a,b Within a row, means with different superscripts differed significantly (*p* < 0.05).

## Data Availability

Data supporting the findings of this study are available from the corresponding author upon reasonable request.

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
