# Peer review of "Morphophysiological Assessment of the Cervix during the Reproductive Cycle and Early Pregnancy in Does Using Computed Tomography and Oxytocin Receptor Immunohistochemistry"

_animals, 2024, doi:10.3390/ani14192793_

Round 1

Reviewer 1 Report

Comments and Suggestions for Authors

The MS entitled “Comprehensive analysis of cervical morphophysiology...” by Kanthawat et al compared the cervical morphophysiology of three stages including follicular, luteal and early pregnant stages based on the whole female reproductive tracts, which is essential for advancing reproductive management and intervention strategies in veterinary practice.

  General comment

The authors analyzed the cervical morphology of folds by CT scanning, collagen, muscular and mucosubstances by staining, gene and protein abundances of OXTR by IHC, RT-qPCR and WB. The MS is of certain novelty and basically well carried out, however, the cervical tissue is heterogeneous and the OXTR expression is cell specific, so how to collect the tissue samples for the RT-PCR and WB, which might affect the result; in addition, several issues listed below need to be elucidated and improved before publication in this journal. 

 Specific comment

L140 The section preparation for the Masson trichrome and Alcian blue staining need to be elucidate first.

L181 Regarding to the RT-qPCR, more details needed based on MIQE, such as Number within each group, Sequence accession number, Amplicon length, Reaction volume and amount of cDNA/DNA, reaction system, Temperature and time

L259 Check Grade of corresponding figure

L284 explain the meaning of different color in the figure 

L319 higher OXTR expression was demonstrated in ...this sentence is not complete

L335 Figure 7: the figure and table indicate the same thing that is immunohistochemical result, so no need to repeat. In addition, no c in the table

L338 How to calculate the relative expression of OXTR protein? Blots of the internal reference need to be shown simultaneously. In addition, do not need to repeat the data in the text as in the figure

L351 Regarding to the RT-qPCR, whats the number in each group? Or whats the biological and technical repeats for each group?

Discussion part: L508- 513, a little bit repetition about the discrepancy of mRNA and protein. Are there any references indicating the post-transcriptional regulatory? please cite them to explain

Comments on the Quality of English Language

Moderate editing of English language required.

Reviewer 2 Report

Comments and Suggestions for Authors

The paper comprehensively studied the morphophysiology of the cervix and the dynamic regulation of oxytocin receptor (OXTR) expression in doe goats, gives better understanding of reproductive management strategies, such as artificial insemination techniques and reproductive health management, and has potential implications for improving reproductive success in practice. On the whole, the abstract is informative, the introduction fully explains the reasons for carrying out the research, the materials and methods is adequately described, and the conclusion is supported by the presented data.

The following modifications are suggested.

Title: The title is descriptive but could be made more concise and general. Consider simplifying it to "Morphophysiological Analysis of Doe Goat Cervix Across Reproductive Stages."

Line18 Gene names ("OXTR ") should be written in italics, please check the full text and modify.

Line 40-43 The transition may be smoother with a presentation from the general role of the cervical canal to its specific role in goats.

Line 45-46 "Cervical rings that are misaligned" could be rephrased to "Misaligned cervical rings" for better flow.

Line 267,274,281,303,320,335,336,347 P values should be formatted according to the rules of the journal to enhance readability.

Line 137 Incomplete labeling of figure note (B).

Line 258-261 Lowercase letters in the figure notes and uppercase letters in the pictures, please be consistent.

Line 227-228 Statistical methods are mentioned, make it clear which one-way ANOVA is used and which is the LSD post-hoc test.

Line 325 (C), (F), (I), (L) Picture scale display is incomplete, replace with correct picture.

Line 337 Please provide the results of the internal reference assay for the WB experiment, and the complete WB image should contain the marker, target protein, and internal reference protein.

Line 405-412 The discussion on Masson trichrome and Alcian blue staining results is insightful but could benefit from a deeper analysis connecting these findings to known physiological processes or previous studies.

Line 536-604 The reference format needs to be consistent with the journal's required format, with some reference years bolded and some not.

"Reference source not found" appears in many places throughout the text, please check and modify it.

Comments on the Quality of English Language

The author demonstrates relatively solid basic skills in English, with clear sentence structures and proper word choice. In particular, the use of professional terminology in the article is appropriate, indicating that the author has a good grasp of specialized knowledge. Additionally, long sentences are skillfully employed to present complex concepts, which is commendable.

Reviewer 3 Report

Comments and Suggestions for Authors

The manuscript entitled "Comprehensive analysis of cervical morphophysiology in doe goats using CT scan, Masson trichrome, Alcian blue staining, and Oxytocin receptor" aimed to analyze the morphophysiology and oxytocin receptor (OXTR) expression in the cervix of doe goats during various reproductive stages. The theme is of interest for veterinary medicine. The manuscript is well structured and original in content. I suggest some amendments for improving the scientific soundness of the manuscript, as follow:

Line 102 - CRL: add this acronym after its respective expression too

Line 175 - add information about the tissues used as positive control

Table 1 - please revise the numbers of 1st and 2nd fold. In both cases, there does not seem to be any differences between follicular and luteal stages and between luteal and pregnant stages

Fig 4 and 5 - What means "epi", "sub" and "m". Add the information in subtitles

Lines 297-298 - "In contract, no....in this study". Delete this statement. It's obvious.

Fig 5 - Delete B, D and F. Show just the positive results.

Fig. 6 - Figures are very small. In addition, delete the figures with negative reaction.

Reviewer 4 Report

Comments and Suggestions for Authors

The study intended to bring information in the gross structure of goat cervix using an innovative approach. The results fill a gap on information in the species that was required and therefore is a valuable contribution. However, authors need to bring precision in the description of methods and results. Discussion need a better organization and authors need to acknowledge that the design was not intended to elucidate the regulation of cervical function, also, inferences need to be supported by the experimental data avoiding excessive speculation; conclusions, also should be an output of the study avoiding overreaching statements. Detailed suggestions and observations are added in the attached file.

Comments on the Quality of English Language

English language needs a moderate edition.

Round 2

Reviewer 1 Report

Comments and Suggestions for Authors Regarding to the protein expression, blots of the internal reference from the same membrane are needed to recalculate the relative expression for the interest protein. It’s not appropriate to normalized it to the total protein. Otherwise, I recommend deleting this result. L211: The writing needs to be improved based on MIQE guideline

Reviewer 4 Report

Comments and Suggestions for Authors

This version is better than the original one; the first table and figures are well designed. Information given in Table 2 is inconsistent with the one in figures, and so, Figure 7. There is information that require precision and details are given in the attached file. However, the Discussion section need a major review not only by discussing this information against the prevalent view in ruminants, but also avoiding conceptual pitfalls. Also it needs an integration of the information gather in this study in goats with the present conceptual framework in ruminants. The discussion needs to be concise and with little room for speculations, excepting tentative hypothesis but well supported by evidence, even obtained in other species. 

Comments on the Quality of English Language

Writing needs a moderate edition.
